# Prematurity—Risk Factor or Coincidence in Congenital Glaucoma?

**DOI:** 10.3390/medicina58030334

**Published:** 2022-02-22

**Authors:** Christiana D. M. Dragosloveanu, Vasile Potop, Valeria Coviltir, Valentin Dinu, Mihai Păsărică, Irina L. Ducan, Călina Maier, Şerban Dragosloveanu

**Affiliations:** 1Department of Ophthalmology, “Carol Davila” University of Medicine and Pharmacy, 050474 Bucharest, Romania; potopvasile@gmail.com (V.P.); valeriacoviltir@yahoo.com (V.C.); dinu_valentin2000@yahoo.com (V.D.); m.pasarica@yahoo.com (M.P.); irina.ducan@yahoo.com (I.L.D.); 2Clinical and Emergency Ophthalmology Hospital Bucharest, 010464 Bucharest, Romania; 3Department of Obstetrics and Gynecology, “Carol Davila” University of Medicine and Pharmacy, 050474 Bucharest, Romania; calinamaier@yahoo.com; 4Department of Orthopedics, “Carol Davila” University of Medicine and Pharmacy, 050474 Bucharest, Romania; serbandrago@gmail.com

**Keywords:** congenital glaucoma, risk factors, prematurity

## Abstract

*Background and* *Objective:* To correlate the intraocular pressure with the postconceptional age and identify a statistically significant connection between congenital glaucoma and prematurity. *Materials and* *Methods:* The current paper is a retrospective, comparative, case-control study. Data collection featured maternal age, gestational age at birth, birth weight, and intraocular ocular pressure (IOP) measurements. *Results:* Forty-two eyes of 21 children underwent examination. The participants were assigned into two groups. The Preterm-Glaucoma (PG) group included eight preterm-born children diagnosed with glaucoma, whereas the Preterm (P) group was comprised of premature newborns without the aforementioned condition. There was no statistically relevant difference in birth weight (*p* = 0.691078) nor in mean gestational age (*p* = 0.752623) between the two groups. The mean IOP in the PG group was 23.813 ± 4.5493, whereas in the P group, it ranged around 13.231 ± 1.0699, *p* < 0.0001. Using mixed-effects models, we obtained a reduction in IOP of 0.45 mmHg per week in the first month of life. A further weekly reduction of 0.36 mmHg was achieved in the next two months. *Conclusions:* The mean IOP of prematurely born children decreased with age. Our findings correlate with previously conducted studies, however, the drop in IOP values exceeded any data published so far. We found no correlation between prematurity and the incidence of congenital glaucoma.

## 1. Introduction

Gestational age is a universal variable used to describe fetal maturation. It is defined by the number of weeks that have passed from the first day of the mother’s last menstrual cycle up until delivery [1].

An ‘at term’ infant is born alive after 37 weeks of gestation. Any birth taking place before this date will result in a preterm-born child [2].

Preterm births are classified in:Extremely preterm: <28 weeks.Very preterm: 28–32 weeks.Moderate/late preterm: 32–37 weeks [3].

Every year, approximately 15 million children are born prematurely (more than 1 in 10 newborns) [3], and over one million of these infants die because of preterm-birth-related complications [4]. Thus, on a global scale, prematurity represents the leading cause of mortality in children under five years of age. Many of the survivors will sustain lifelong visual and hearing disabilities.

Although more than 60% of preterm births occur in Africa and South Asia, prematurity is truly a worldwide issue. Nevertheless, low-income countries have higher preterm birth rates when compared to more developed ones (12% versus only 9%), and within each country, poverty is one of the leading factors to influence prematurity [5].

Preterm labor is now regarded as a syndrome initiated by multiple mechanisms, including infection or inflammation, uteroplacental ischemia or hemorrhage, uterine overdistension, stress, and other immunologically mediated processes [6]. Nevertheless, a precise mechanism cannot be established in most cases.

Congenital glaucoma is an uncommon ocular condition and one of the main causes of blindness in children. It commonly occurs among communities that encourage consanguinity [7] and may be classified as primary (without any ocular or systemic developmental anomalies) or secondary congenital glaucoma (accompanied by other pathologies) [8,9]. Regardless of its type, the disease affects one’s normal development from early childhood until adulthood, thus having a significant impact on quality of life [8,9,10].

A surgical approach is the only feasible treatment and should be carried out as soon as possible after a diagnosis has been established. However, the available procedures are some of the most difficult and complex in glaucoma surgery [11,12]. Children have a higher post-surgical risk of failure and complications, mostly due to buphthalmia and aggressive healing [11,12].

In this regard, other systemic abnormalities that may accompany preterm births only add a further burden to the surgical and postoperative strategy. One study carried out by Zertsalova et al. stated that the majority of congenital glaucoma patients were, in fact, preterm-born [13]. This seems to be in agreement with the etiological process proposed by Fang Ko in which the immature angle found in congenital glaucoma comes to a developmental standstill at a certain point in the third trimester of pregnancy [14].

However, these findings contradict the trend that we noticed in our practice, hence we decided to further study this dissimilarity.

## 2. Patients and Methods

### 2.1. Study Design and Patient Recruitment

In the already published retrospective studies [7,8] conducted between 2010–2020, we observed that, from a total of 45 patients with primary congenital glaucoma, 8 patients were born preterm (group PG).

We enrolled another 13 prematurely born infants without congenital glaucoma, who required at least 3 months of ophthalmologic follow-ups for retinopathy of prematurity (ROP) screening (group P).

The study received approval from the local ethics committee.

Written consent was obtained from all subjects’ parents and/or legal guardians after receiving thorough explanations for possible benefits and potential risks.

### 2.2. Inclusion and Exclusion Criteria

Inclusion criteria encompassed the following: Preterm-born children, with (PG group) or without (P group) primary congenital glaucoma, aged 1 day to 3 years, and followed-up for a minimum of 1 year [7,8].

The sought-out exclusion criteria were secondary congenital glaucomas and any ocular or facial traumas, as stated in the already published studies [7,8].

### 2.3. Investigation and Examinations

We assessed the general ocular appearance and the patients’ visual behavior. All examinations were performed under low-dose inhalation anesthesia by the same team. The intraocular pressure (IOP) was measured using a Perkins tonometer in both groups. The PG group underwent further corneal diameter and axial length measurements, whereas the P group was assessed for retinopathy of prematurity via indirect ophthalmoscopy. Gonioscopy could not be properly performed (technical microscopic issues) in all cases, hence we did not include it in our study.

All patients in the PG group underwent surgery for congenital glaucoma the following day after examination [7,15], while subjects in the P group who had associated ROP started treatment for this disorder.

### 2.4. Follow-Up

Subjects in the PG group underwent measurements at the time of their referral for congenital glaucoma suspicion, with no prior data available. In the P group, the IOP was assessed at birth, with follow-up visits after one and three months, respectively.

### 2.5. Outcomes

The aim of this study is to identify a statistically significant connection between congenital glaucoma and prematurity, as well as to correlate the variation of intraocular pressure with postconceptional age.

## 3. Results

All the cases were distributed into two groups: Eight premature patients diagnosed with congenital glaucoma (PG) and 13 preterm subjects without congenital glaucoma (P) (Table 1).

The number of weeks of gestation ranged between 28 and 35 weeks in both groups. In the PG group, the mean value was 31.625 ± 2.56 weeks, whereas in the P group, it arrayed around 31.308 ± 1.974 weeks (Table 2).

The difference between weeks of gestation in the two groups is without statistical significance, *p* = 0.75, >0.1 (*t*-test for Equality of Means, equal variances assumed).

The birth weight was higher in the PG group (1659.375 ± 420.53 g) than in P group (1586.923 ± 386.83 g), with a *t*-test (*t*-test for Equality of Means, equal variances assumed) *p* = 0.691 (>0.1), however, without being statistically relevant (Table 3).

In what concerns sex dispersion, 11 boys and 10 girls were assigned to the two groups: Three boys (37.5%) and five girls (62.5%) in the PG group, respectively, eight boys (61.54%) and five girls (38.46%) in the P group. (Figure 1).

Comparing the values in each group, we have a Fischer test of *p* > 0.1, thus without any statistically significant differences.

As for the delivery process at birth, there were five natural births and three c-sections in the PG group, alongside nine natural births and four c-sections in the P group. *p* values were not statistically significant between groups > 0.99 (Fischer’s Exact Test). The cause that led to premature birth was specified in each subject’s hospitalization form (Table 4):

There were six children with other associated disorders (Table 5).

Maternal age at birth was assessed as well, with a mean of 29.625 ± 6.631 years in the PG group (ranging from 16 to 39 years) and around 31.076 ± 5.361 years in the P group (19 to 41 years). However, differences were not statistically significant, with a *p* = 0.587, >0.1. (*t*-test for Equality of Means, equal variances assumed).

Subjects in the PG group underwent measurements at the time of their referral for congenital glaucoma suspicion, with no prior data available. Thus, the variable used for time positioning of the IOP values was the mean age at the moment of diagnosis (expressed in months) = 3.875 ± 1.125 months.

In the P group, the IOP was assessed at birth, with follow-up visits after one and three months, respectively. To evaluate the differences in IOP between the two groups, we used the mean IOP in the PG group and the mean IOP at three months from the P group. In order to ensure a relevant outcome, the mean ages of the groups were tested for any significant differences (3.875000 ± 1.125992 months in the PG group versus 3.000000 ± 0.000000 months in the P subgroup, *p* = 0.063924,), employing an independent *t*-test. As the mean age was akin, without any statistically important dissimilarities between the two groups, we proceeded to analyzing the mean IOPs.

Therefore, the mean IOP in the PG group was roughly 23.813 ± 4.549 mmHg and raging around 13.231 ± 1.069 mmHg in the P group. The *t*-test *p* value turned out <0.001, revealing that the difference between the two groups was statistically significant (Table 6).

The IOPs in the P group were measured at birth, after one and three months, respectively. The mean values were then compared (Table 7).

IOP RE + LE (78 measurements): 14.858 ± 1.985, range = (11–21).IOP RE (39 values): 14.743 ± 1.697, range = (11–18).IOP LE (39 values): 14.974 ± 2.253, range = (11–21).

Two values (2.56%) have been bigger than 20 (21 mmHg) at birth, decreasing in the next period.

Dividing IOP in percentiles (P10–P90) by age, in P group, we obtain (Table 8):

Applying a statistical mixed-effect calculation, we obtain an IOP reduction for both eyes with:1.807692 (*p*-value = 0.000010, 95% CI, −2.487959, −1.127426) between birth and one month,3.346154 (*p*-value < 0.001, 95% CI, −3.957952, −2.734356) between birth and three months,

For RE with:1.307692 (*p*-value = 0.001138, 95% CI, −1.988213, −0.627172), birth—one month2.923077(*p*-value < 0.001, 95% CI, −3.360331, −2.485823), birth—three months

And for LE:2.307692 (*p*-value = 0.001020, 95% CI, −3.491783, −1.123602), birth—one month3.769231(*p*-value = 0.000009, 95% CI, −4.924259, −2.614202), birth—three months

Thus, an important decrease in IOP values can be noticed over time. This observation is statistically significant (*p* < 0.001).

## 4. Discussion

All the measurements were carried out under low-dose inhalation anesthesia, thus being less uncomfortable for the infant and more exact for the study. In our experience, using anesthetic eye drops and any kind of speculums while the infant is actively moving leads to falsely elevated IOP values (with up to 4 mmHg). This finding was also suggested by one study [16] using an Alfonso eyelid speculum.

There is currently only a handful of studies addressing this issue, all of which have a shorter follow-up period. To our knowledge, this is the longest study assessing the IOP trend in prematurely born children. IOP measurements were performed at birth and after one and three months, respectively. The mixed-effect model was employed in order to evaluate the IOP shift, which turned out to have a statistically significant decrease with age.

Ricci [17] evaluated the IOP of 20 preterm-born children on a weekly basis, from birth up to one month. The study did not find a statistically relevant correlation between aging and a drop in IOP values, attributing this result to the fact that the aqueous drainage system fully develops later on in life.

Ng et al. [18] measured the IOP of 104 prematurely born infants at 1, 4, 6, 8, and 10 weeks after birth, also failing to obtain a correlation between postnatal age and a decrease in IOP. In this study, the gestational age ranged around 29.8 weeks, whereas the mean birth weight was approximately 1208 g. In comparison, the mean gestational age in our study was 31.625 ± 2.559994 weeks in the PG group and 31.307692 ± 1.974192 weeks in P group (*p* = 0.752623), variables having a higher value, but without statistical importance. The birth weight values also exceeded those of Ng et co.: 1659.375 ± 420.530596 g in the PG group, 1659.375000 ± 420.530596 g in the P group (*p* = 0.691078). In the aforementioned study, a mean IOP reduction of 0.11 mmHg (*p* < 0.001) was observed with each week of postconceptional age. In contrast, we noticed a reduction of 0.45 mmHg per week in the first month of life, and a mean reduction of 0.36 mmHg per week in the following two months.

Lindenmeyer et al. [19] examined 50 preterm infants, with weekly follow-ups for one month. The mean gestational age was 29.7 ± 1.6 weeks and the mean birth weight was 1,127.7 ± 222.7 g. The mean intraocular pressure was 14.9 ± 4.5 mmHg in both eyes. A mean IOP reduction of 0.29 mmHg was obtained for each postconceptional week that had passed (*p* = 0.047; 95% CI: −0.58 to −0.0035, decreasing from 16.3 mmHg (10.52–22.16) at 26.3 weeks to 13.1 mmHg (7.28–18.92) at 37.6 weeks [19]. Thus, the drop in IOP values was similar to our study but nonetheless on the lower side.

It should also be stated that all of the above-mentioned studies do not take into account other variables, such as sex distribution, maternal age at birth, the type of delivery procedures, causes of premature birth, and other associated disorders.

On a separate note, our study was comprised of two groups, and in one of them the preterm infants were diagnosed with congenital glaucoma (the PG group).

In regards to this aspect, the current literature is still deficient. In one study, Thiagarajah et al. [20], stated that the incidence of congenital glaucoma in the preterm neonate population (2%) is significantly higher than the incidence commonly found in the general population and that premature newborns are at a greater risk of developing this abnormality.

Senthil et al. [21] assessed 15,000 prematurely born infants, out of which 3000 children had ROP. Eighty-seven eyes of 57 premature children had glaucoma—five eyes (5.7%) of three children in the entire ROP cohort had coexisting congenital glaucoma (before any ROP intervention). Three eyes of two children had primary congenital glaucoma and two eyes of one child had glaucoma with microspherophakia.

These studies suggest that there can be a correlation between prematurity and congenital glaucoma, justifying our search in the current paper. However, comparing the mean IOP values in our groups, 23.813 ± 4.5493 in PG group and 13.231 ± 1.0699 in the P group (values at three months after birth), we obtained a *t*-test *p* value of <0.0001. Thus, obtaining a statistically significant difference between the two values, we can state that, to our knowledge, there is no correlation between prematurity and the incidence of congenital glaucoma in this population.

When discussing limitations, we are aware of the bias induced by the small number of patients encompassed in this start-point article and the necessity of a future, more ample study based on this research idea. Although the number of subjects included in our study may raise questions about the relevance of our statistics, it is of importance to state that most, if not all the congenital glaucoma cases in Romania are referred to the Clinical Emergency Eye Hospital in Bucharest for treatment. To our knowledge, there is no other hospital or private practice to address this illness in our country. Therefore, the current study should be regarded as an overview of the annual trend in congenital glaucoma in communities and regions similar to Romania.

## 5. Conclusions

No correlation was found between preterm birth and the incidence of congenital glaucoma. Furthermore, it appears that in the preterm-born population, the IOP tends to decrease with age.

## Figures and Tables

**Figure 1 medicina-58-00334-f001:**
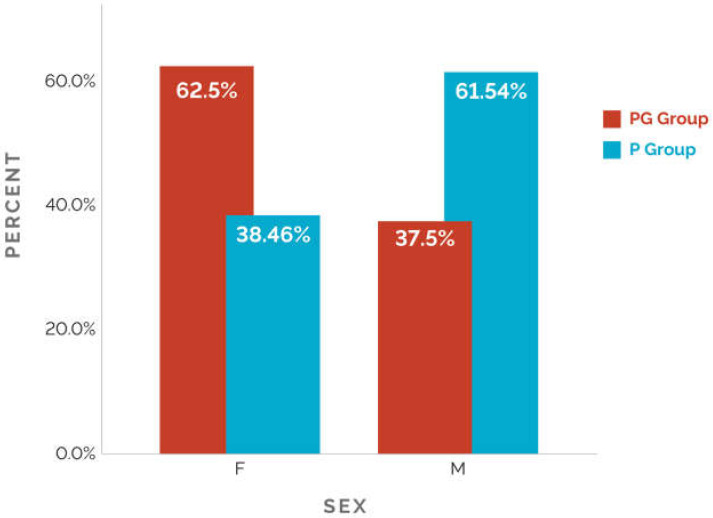
Sex dispersion in both groups.

**Table 1 medicina-58-00334-t001:** The two groups of subjects.

	Frequency	Percent
PG Group	8	38.1
P Group	13	61.9
Total	21	100.0

**Table 2 medicina-58-00334-t002:** Weeks of gestation in both groups.

				Statistic	Std. Error
Weeks of gestation	PG Group	Mean		31.625	0.9051
		95% Confidence Interval for Mean	Lower Bound	29.485	
			Upper Bound	33.765	
		Median		32.000	
		Variance		6.554	
		Std. Deviation		2.5600	
		Minimum		28.0	
		Maximum		35.0	
		Skewness		−0.465	0.752
		Kurtosis		−0.823	1.481
	P Group	Mean		31.308	0.5475
		95% Confidence Interval for Mean	Lower Bound	30.115	
			Upper Bound	32.501	
		Median		31.000	
		Variance		3.897	
		Std. Deviation		1.9742	
		Minimum		28.0	
		Maximum		35.0	
		Skewness		0.262	0.616
		Kurtosis		−0.259	1.191

**Table 3 medicina-58-00334-t003:** Birth weight variables in both groups.

				Statistic	Std. Error
Weight at birth	PG Group	Mean		1659.375	148.6800
		95% Confidence Interval for Mean	Lower Bound	1307.803	
			Upper Bound	2010.947	
		Median		1650.000	
		Variance		176,845.982	
		Std. Deviation		420.5306	
		Minimum		1085.0	
		Maximum		2380.0	
		Skewness		0.358	0.752
		Kurtosis		0.039	1.481
	P Group	Mean		1586.923	107.2882
		95% Confidence Interval for Mean	Lower Bound	1353.162	
			Upper Bound	1820.684	
		Median		1600.000	
		Variance		149,639.744	
		Std. Deviation		386.8330	
		Minimum		1020.0	
		Maximum		2420.0	
		Skewness		0.730	0.616
		Kurtosis		0.387	1.191

**Table 4 medicina-58-00334-t004:** Premature birth causes in each group.

	PG Group	P Group	*p*-Value
	8	13	
abruptio placentae	0/8 (0.0%)	1/13 (7.7%)	0.758 (>0.1)
cervical incompetence	0/8 (0.0%)	1/13 (7.7%)	
preeclampsia	1/8 (12.5%)	1/13 (7.7%)	
domestic violence	0/8 (0.0%)	1/13 (7.7%)	
growth restriction	1/8 (12.5%)	2/13 (15.4%)	
premature rupture of membranes	5/8 (62.5%)	6/13 (46.2%)	
multiple pregnancy	1/8 (12.5%)	1/13 (7.7%)	

**Table 5 medicina-58-00334-t005:** Associated disorders.

		Frequency	Percent
	aniridia (PG group)	1	4.8
	hearing disorder (P group)	1	4.8
	psychomotor retardation (PG group)	1	4.8
	psychomotor retardation (P group)	1	4.8
	retinopathy of prematurity (P group)	1	4.8
	Sturge Weber sdr (PG group)	1	4.8
	Total	6	28.6
Without other disorders		15	71.4
Total		21	100.0

**Table 6 medicina-58-00334-t006:** *p* value for IOP at three months in both groups.

	PG Group	P Group (3 Months)	*p*-Value
	16 values	26 values	
IOP	23.813 ± 4.549	13.231 ± 1.069	<0.0001 *

* *t*-test for Equality of Means, equal variances not assumed. IOP—intraocular pressure.

**Table 7 medicina-58-00334-t007:** Mean IOP values at birth, one month, and three months.

		Patients	Minimum	Maximum	Mean	SD
birth	Age (weeks)	13	28	35	31.307	1.974
	IOP RE	13	14	17	16.153	0.898
	IOP LE	13	15	21	17.000	2.236
1 month	Age (weeks)	13	32	39	35.307	1.974
	IOP RE	13	11	18	14.846	1.675
	IOP LE	13	12	16	14.692	1.315
3 months	Age (weeks)	13	40	47	43.307	1.974
	IOP RE	13	12	15	13.230	0.926
	IOP LE	13	11	15	13.230	1.235

IOP—intraocular pressure, RE—right eye, LE—left eye.

**Table 8 medicina-58-00334-t008:** IOP in percentiles (P10–P90) by age.

Age	Mean IOP	P10–P90 IOP	Min–Max IOP
28	16	15–17	15–17
29	17	16–18	16–18
30	16.5	15–19	15–19
31	17.25	15–21	15–21
32	15.625	14–18	14–18
33	15.25	14–17	14–17
34	16.125	14–21	14–21
35	16.00	14–18	14–18
36	13.83	11–15	11–15
37	13.00	12–14	12–14
38	16.50	16–17	16–17
39	14.00	13–15	13–15
40	13.00	12–14	12–14
41	13.50	13–14	13–14
42	13.333	12–15	12–15
43	14.00	13–15	13–15
44	13.333	12–15	12–15
45	12.00	11–13	11–13
46	13.00	13–13	13–13
47	12.50	11–14	11–14

IOP—intraocular pressure.

## Data Availability

Data is available in the Archives of the Clinical and Emergency Ophthalmology Hospital Bucharest.

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
