# Peer review of "Prematurity—Risk Factor or Coincidence in Congenital Glaucoma?"

_medicina, 2022, doi:10.3390/medicina58030334_

Round 1

Reviewer 1 Report

Some comments enclosed:

1. Spell IOP, PG, and P at first use in the abstract
2. In the introduction, there should be a brief summary of past findings about the risk of congenital glaucoma in premature infants, and a description about why is the current study needed (e.g. to verify prior consistent results or to examine the mixed reports)
3. Since the number of cases included was very small, has the possibility of an insufficient statistical power been considered?
4. Although no prior values could be obtained, unsure if using 3 month-IOP of P group as a comparison to 3.9 month-IOP of PG group is appropriate. Also, to state that prematurity does not contribute to the risk of congenital glaucoma simply based the statistical different in IOP did not seem too convincing.
5. Description of table 7 in the result section is a bit unclear. What is "2 values"?
6. The first paragraph of discussion is a bit confusing as well, might need some work on English editing.
7. If most prior studies have already shown a negative correlation between age and IOP, what is the additional value of the current finding? Better to explain how it adds to/differs from the existing knowledge. 
8. Please work on the limitation section and add a short paragraph of conclusion

Author Response

Thank you for your review. My team and I read your suggestions and tried our best to improve our manuscript.

We added a new paragraph to the introduction in order to explain the reason for our current study. 

We used the 3rd month measurements in the P group especially because it was near the mean age in the PG group, and I personally think that the 0,9 months difference has no statistical importance. 

As for points 5,6 and 7 of your review, these were all due to English misspelling and syntax errors. In this regard, we revised the entire manuscript.

I hope that the revised manuscript is in fact publish-worthy.

Thank you once again,

Kind Regards,

Christiana Dragosloveanu 

Reviewer 2 Report

The authors Christiana DM Dragosloveanu et.al in the article titled “Prematurity risk factor or
coincidence in congenital 2 glaucoma? were interested to study for a potential correlation between
prematurity and glaucoma (increase in the intraocular pressure) developed from birth (congenital).
The authors chose to study two groups of population, one with (PG) and without (P) congenital
glaucoma. The aim of the authors was to show a statistically significant correlation between the
prematurity and congenital glaucoma by conducting analysis on gestation period, weight at birth,
sex dispersion (girls vs boys), and type of birth procedure (natural birth vs c-section). However,
no statistically significant data was observed in the above listed data sets. While, IOP was
measured in both groups, it was observed that the IOP levels significantly decreased with age.
Authors arrived at a conclusion stating that there is no correlation identified between prematurity
and congenital glaucoma. Christiana DM Dragosloveanu et.al did a good work in analyzing the
data set by applying appropriate statistical tools in evaluating the p-value on each of the data sets.
The article is introduced and written well. Authors effort in writing the discussion is to be well
appreciated that indicates the references from other studies. Overall, this article is well written.

The questions that may need attention from the authors are listed below:

1. Sex dispersion chart with corresponding IOP levels in both, the group PG and group P: Fig 1
indicates sex dispersion chart in comparison to both groups. It may be good to have this data
with the corresponding IOP levels. This can provide us a summary chart of IOP vs gender
between both groups tested.

2. Tabulating ages of the parents with cases of prematurity: Line # 123-126 has indicated about
the mothers age at birth that showed no significant difference between both groups.
However, tabulating both parents (mother and Father, if data available) age with the gestation
period could further provide correlation between the age of the parents with the prematurity.

3. As mentioned in the discussion (line # 216) the authors are aware of the small size of the
population in the study. Therefore, can this only be considered as a pilot study providing some
insights on potential correlation since Senthil et.al (reference # 19) reported correlation with
increased number of subjects in the study. Agree?

4. Having a lower n# of population in this study and showing no correlation between prematurity
and congenital glaucoma, what impact will this article make for the future study?

Author Response

Thank you for your review. 

My team and I have revised our manuscript, and tried our best to correct all the English grammar and syntax errors. 

Kind regards,

Christiana Dragosloveanu

Round 2

Reviewer 1 Report

I appreciate the author's efforts to revise the manuscript. However, some of the points raised previously were not addressed completely in the text (with only response provided), and there are a few more things to be noted:

  1. The study limitations should be discussed (I believe the current version is too brief and does not really say much)
  2. The author's response stated "I personally think that the 0,9 months difference has no statistical importance." Since the author said so, it will be better to show the statistics in both the response letter and add this point to the result/discussion to strengthen their point.
  3. While extensive English editing has been done, I noticed there wasn't much change in the content of the text in accordance with the reviewers' comments. Please also state in the response letter the changes you made to the manuscript based on the prior comments.

Author Response

Thank you once again for the time invested in reviewing our study. 

I will try to briefly answer all of the requests stated in the two reviews starting with the former. 

  1. Spell IOP, PG, and P at first use in the abstract - This comment was addressed in lines 16,17 and 18 of the resubmitted manuscript.  
  2. In the introduction, there should be a brief summary of past findings about the risk of congenital glaucoma in premature infants, and a description about why is the current study needed (e.g. to verify prior consistent results or to examine the mixed reports) - This comment was seen to in lines 64 -71 of the manuscript.
  3. Since the number of cases included was very small, has the possibility of an insufficient statistical power been considered?  - As for rare diseases this is a possibility; the problem has been addressed in the limitation section of the text. As stated there, our two groups are representative of the annual trend in congenital glaucoma in communities and regions similar to Romania.
  4. Although no prior values could be obtained, unsure if using 3 month-IOP of the P group as a comparison to 3.9 month-IOP of PG group is appropriate. Also, to state that prematurity does not contribute to the risk of congenital glaucoma simply based the statistical different in IOP did not seem too convincing. - It was our mistake not to insert the statistics in the manuscript, thinking the age is obviously similar. We apologize for that. The revised version now contains the p-value of the independent t-test used for the mean age in the two groups, which is 0.063924. As this study does not intend to address the etiology nor the anatomopathological cause of congenital glaucoma, for this would require biopsies, we solely relied on the data obtained from our clinical examinations. This is one possible explanation for what we noticed in our practice.
  5. Description of table 7 in the result section is a bit unclear. What is "2 values"?  - The description was misspelled and left uncorrected. we revised that as well.
  6. The first paragraph of discussion is a bit confusing as well, might need some work on English editing. - The English was edited, if the paragraph is still confusing please let us know in what aspect. 
  7. If most prior studies have already shown a negative correlation between age and IOP, what is the additional value of the current finding? Better to explain how it adds to/differs from the existing knowledge. - This was a translation mistake, as in Romanian "negative correlation" has the exact opposite meaning that it has in English. This is now fixed.
  8. Please work on the limitation section and add a short paragraph of the conclusion - The paragraph has been added in the revised manuscript.
  9. The study limitations should be discussed (I believe the current version is too brief and does not really say much) - This issue has been addressed in the second revised version.
  10. Points 2 and 3 of the second review encompass part of the first review and have been addressed in the explanations offered above.

Thank you,

Best regards, 

Christiana Dragosloveanu

Round 3

Reviewer 1 Report

I thank the authors for their efforts on revising the manuscript. I have no further comments.